# TOPOLOGY-INFORMED GRAPH TRANSFORMER

## ABSTRACT

Transformers have revolutionized performance in Natural Language Processing and Vision, paving the way for their integration with Graph Neural Networks (GNNs). One key challenge in enhancing graph transformers is strengthening the discriminative power of distinguishing isomorphisms of graphs, which plays a crucial role in boosting their predictive performances. To address this challenge, we introduce 'Topology-Informed Graph Transformer (TIGT)', a novel transformer enhancing both discriminative power in detecting graph isomorphisms and the overall performance of Graph Transformers. TIGT consists of four components: A topological positional embedding layer using non-isomorphic universal covers based on cyclic subgraphs of graphs to ensure unique graph representation: A dual-path message-passing layer to explicitly encode topological characteristics throughout the encoder layers: A global attention mechanism: And a graph information layer to recalibrate channel-wise graph features for better feature representation. TIGT outperforms previous Graph Transformers in classifying synthetic dataset aimed at distinguishing isomorphism classes of graphs. Additionally, mathematical analysis and empirical evaluations highlight our model's competitive edge over state-of-the-art Graph Transformers across various benchmark datasets.

## 1 INTRODUCTION

Transformers have achieved remarkable success in domains such as Natural Language Processing (Vaswani et al., 2023) and Computer Vision (Dosovitskiy et al., 2021). Motivated by their prowess, researchers have applied them to the field of Graph Neural Networks (GNNs). They aimed to surmount the limitations of Message-Passing Neural Networks (MPNNs), which are a subset of GNNs, facing challenges such as over-smoothing (Oono & Suzuki, 2021), over-squashing (Alon & Yahav, 2021), and restricted expressive power (Xu et al., 2019; Morris et al., 2021). An exemplary application of the integration of Transformers into the GNNs field is the Graph Transformer. Multi-head attention mechanism of Transformers is applied to each node in the graph treating the entire set of nodes as if they are fully connected or treating the set of nodes if they are connected by edges. These approaches often come with a low inductive bias, making them prone to over-fitting. Consequently, several implementations blend Graph Transformers with or without MPNNs, yielding promising outcomes (Yang et al., 2021; Ying et al., 2021; Dwivedi & Bresson, 2021; Chen et al., 2022; Hussain et al., 2022; Zhang et al., 2023; Ma et al., 2023; Rampášek et al., 2023; Kong et al., 2023; Zhang et al., 2022).

While Graph Transformers have marked considerable advancements, enhancing Graph Transformers through strengthening the discriminative power of distinguishing isomorphisms of graphs is remaining challenge, which plays a crucial role in boosting their graph-level predictive performances. Previous research has explored various techniques to address the limitations of discriminative power. For instance, studies based on MPNN have enhanced node attributes using high-dimensional complexes, persistent homological techniques, and recurring subgraph structures (Carrière et al., 2020; Bodnar et al., 2021b; Bouritsas et al., 2021; Wijesinghe & Wang, 2021; Bevilacqua et al., 2022; Horn et al., 2022; Choi et al., 2023). Similarly, recent research on Graph Transformers has investigated the use of positional encoding grounded in random walk strategies, Laplacian PE, node degree centrality, and shortest path distance to address these limitations. Furthermore, structure encoding

based on substructure similarity has been introduced to amplify the inductive biases inherent in the Transformer.

This paper introduces a Topology-Informed Graph Transformer(TIGT), which embeds sophisticated topological inductive biases to augment the model's expressive power and predictive efficacy. Before the Transformer layer, each node attribute is integrated with a topological positional embedding layer based on the differences of universal covers obtained from the original graph structure and collections of unions of cyclic subgraphs, the topological invariants of which contains their first homological invariants. In companion with the novel positional embedding layer, we explicitly encode cyclic subgraphs in the dual-path message passing layer and incorporate channel-wise graph information in the graph information layer. These are combined with global attention across all Graph Transformer layers, drawing inspiration from Choi et al. (2023) and Rampášek et al. (2023). As a result, the TIGT layer can concatenate hidden representations from the dual-path message passing layer, combining information of original structure and cyclic subgraphs, global attention layer, and graph information layer to preserve both topological information and graph-level information in each layer. Specifically, the dual-path message passing layer enables overcoming the limitations of positional encoding and structural encoding to increase expressive power when the number of layers increases. We justify the proposed model's expressive power based on the theory of covering space. Furthermore, we perform experiment in synthetic datasets aimed at distinguishing isomorphism classes of graphs and benchmark datasets aimed at demonstrating the state-of-the-art of competitive predictive performance of the proposed model.

Our main contributions can be summarized as follows: (i) Theoretical justification of expressive powers of TIGT and its comparison with other Graph Transformers by utilizing the theory of covering spaces, comparison of Euler characteristic formulae of graphs and their subgraphs, and the geometric rate of convergence of Markov operators over finite graphs to stationary distributions. (ii) Novel positional embedding layer based on the MPNNs and simple architectures to enrich topological information in each Graph Transformer layer (iii) Outperformance shown in processing synthetic dataset to assess the expressive power of GNNs (iv) State-of-art or competitive results, especially in the large graph-level benchmarks.

## 2 PRELIMINARY

**Message passing neural networks** MPNNs have demonstrated proficiency in acquiring vector representations of graphs by handling local information based on the connectivity between nodes among other types of GNNs such as Graph Convolution Network (GCN), Graph Attention Network (GAT) (Veličković et al., 2018), Graph Isomorphism Network (GIN) (Xu et al., 2019) and Residual Graph ConvNets (GatedGCN) (Bresson & Laurent, 2018). We denote $\text{MPNN}^l$ when it has a composition of $l$ neighborhood aggregatin layers. Each $l$-th layer $H^{(l)}$ of the network constructs hidden node attributes of dimension $k_l$, denoted as $h_v^{(l)}$, using the following composition of functions:

$$\begin{cases} h_v^{(l)} := \text{COMBINE}^{(l)} \left( h_v^{(l-1)}, \text{AGGREGATE}_v^{(l)} \left( \left\{ \left\{ h_u^{(l-1)} \mid \substack{u \in V(G), u \neq v \\ (u,v) \in E(G)} \right\} \right\} \right) \right) \\ h_v^{(0)} := X_v \end{cases}$$

where $X_v$ is the initial node attribute at $v$. Let $M_v^{(l)}$ be the collection of all multisets of $k_{l-1}$-dimensional real vectors with $\deg v$ elements counting multiplicities.

$$\text{AGGREGATE}_v^{(l)} : M_v^{(l)} \to \mathbb{R}^{k_l'}$$

is a set theoretic function of $k_l'$-dimensional real vectors, and the combination function

$$\text{COMBINE}^{(l)} : \mathbb{R}^{k_{l-1}+k_l'} \to \mathbb{R}^{k_l}$$

is a set theoretic function combining the attribute $h_v^{l-1}$ and the image of $\text{AGGREGATE}_v^{(l)}$.

Let $M^{(L)}$ be the collection of all multisets of $k_L$-dimensional vectors with $\#V(G)$ elements. Let

$$\text{READOUT} : M^{(L)} \to \mathbb{R}^K$$

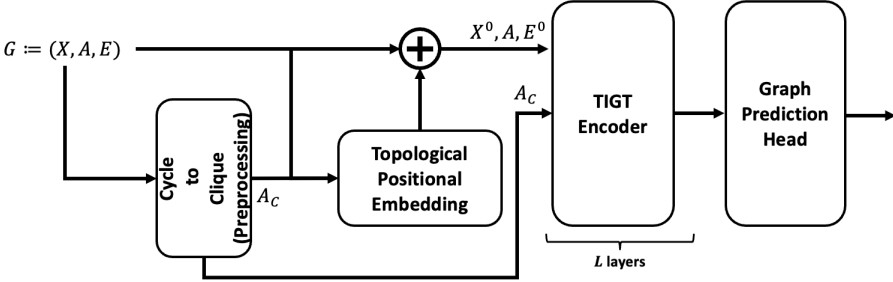

Figure 1: Overall Architecture of TIGT.

be the graph readout function of $K$-dimensional real vectors defined over the multiset $M^{(L)}$. Then the $K$-dimensional vector representation of $G$, denoted as $h_G$, is given by

$$h_G := \text{READOUT}\left(\{\{h_v^{(l)} \mid v \in V(G)\}\}\right)$$

**Clique adjacency matrix** The clique adjacency matrix, proposed by Choi et al. (2023), is a matrix that represents bases of cycles in a graph in a form analogous to the adjacency matrix, enabling its processing within GNNs. Extracting bases of cycles results in incorporating a topological property equivalent to the first homological invariants of graphs (Paton, 1969). The set of cyclic subgraphs of $G$ which forms the basis of the cycle space (or the first homology group) of $G$ is defined as the cycle basis $\mathcal{B}_G$. The clique adjacency matrix, $A_C$, is the adjacency matrix of the union of $\#\mathcal{B}_G$ complete subgraphs, each obtained from adding all possible edges among the set of nodes of each basis element $B \in \mathcal{B}_G$. Explicitly, the matrix $A_C := \{a_{u,v}^C\}_{u,v \in V(G)}$ is given by

$$a_{u,v}^C := \begin{cases} 1 & \text{if } \exists\, B \in \mathcal{B}_G \text{ cyclic s.t. } u, v \in V(B) \\ 0 & \text{otherwise} \end{cases}$$

We note that it is also possible to construct bounded clique adjacency matrices, analogously obtained from sub-bases of cycles comprised of bounded number of nodes.

## 3 TOPOLOGY-INFORMED GRAPH TRANSFORMER(TIGT)

In this section, we introduce overall TIGT architecture. The overall architecture of our model is illustrated in Figure 1.Suppose we are given the graph $G := (V, E)$. It can be represented by four types of matrices to use input of TIGT; a node feature matrix $X \in \mathbb{R}^{n \times k_X}$, an adjacency matrix $A \in \mathbb{R}^{n \times n}$, a clique adjacency matrix $A_c \in \mathbb{R}^{n \times n}$ and an edge feature matrix $E \in \mathbb{R}^{n \times k_E}$. Note that $n$ is the number of node, $k_X$ is node feature dimension and $k_E$ is edge feature dimension. The clique adjacency matrices are obtained from same process in previous research (Choi et al., 2023). For clarity and conciseness in our presentation, we have omitted the details pertaining to the normalization layer and the residual connection. We note that some of the mathematical notations used in explaining the model design and details of model structures conform to those shown in (Rampášek et al., 2023).

### 3.1 TOPOLOGICAL POSITIONAL EMBEDDING LAYER

Most of previous researches of Graph Transformers use positional embeddings based on parameters such as node distance, random walk, or structural similarities, in order to adequately capture the nuances of graph structures. Diverging from this typical approach, we propose a novel method for obtaining learnable positional encodings by leveraging MPNNs. This approach aims to enhance the discriminative power with respect to isomorphism classes of graphs, drawing inspiration from Cy2C-GNNs (Choi et al., 2023). First, we use any MPNNs to obtained hidden attribute from original

graph and new graph structure with clique adjacency matrix as follows:

$$h_A = \text{MPNN}(X, A), \qquad h_A \in R^{n \times k}$$
$$h_{A_C} = \text{MPNN}(X, A_C), \quad h_{A_C} \in R^{n \times k}$$
$$h = [h_A \quad h_{A_C}], \qquad h \in R^{n \times k \times 2}$$

where $X$ represents the node embedding tensor from the embedding layers, and $[\ \ ]$ denotes the process of stacking two hidden representations. It's important to note that MPNN for adjacency matrix and clique adjacency matrix share weights, and the number of layers of MPNNs are not a constraint. Then the node features are updated along with topological positional attributes, as shown below:

$$X_i^0 = X_i + \text{SUM}(\text{Activation}(h_i \odot \theta_{pe})), \quad X^0 \in R^{n \times k}$$

where $i$ is the node index in graph and $\theta_{pe} \in R^{1 \times k \times 2}$ represents the learnable parameters that are utilized to integrate features from two different universal cover. The SUM operation performs a sum of the hidden features $h_A$ and $h_{A_c}$ by summing over the last dimensions. For the Activation function, in this study, we use hyperbolic tangent function to bound the value of positional information. Regardless of the presence or absence of edge attributes, we do not use any existing edge attributes in this layer. The main objective of this layer is to enrich node features by adding topological information by combining the two universal covers. The results $X^0$ will be subsequently fed into the encoder layers of TIGT.

## 3.2 Encoder layer of TIGT

The Encoder layer of the TIGT is based on the three components: A Dual-path message passing layer: A global attention layer: And a graph information layer. For the input to the Encoder layer, the transformed input feature $X^{l-1}$, along with $A$, $A_c$, and $E^{l-1}$ is utilized, where $l$ is the encoder layer number in TIGT.

**Dual-path MPNNs**   Hidden representations $X^{l-1}$, sourced from the preceding layer, paired with the adjacency matrix and clique adjacency matrix, are processed through a dual-path message passing layer as follows:

$$X_{MPNN,A}^l = \text{MPNN}_A(X^{l-1}, E^{l-1}, A), \quad X_{MPNN,A}^l \in R^{n \times k}$$
$$X_{MPNN,A_C}^l = \text{MPNN}_{A_C}(X^{l-1}, A_C), \quad X_{MPNN,A_C}^l \in R^{n \times k}$$

**Global attention layer**   To capture global relationship of each node, we apply multi-head attention of vanilla Transformer as follows:

$$X_{MHA}^l = \text{MHA}(X^{l-1}), \quad X_{MHA}^l \in R^{n \times k}$$

where MHA is multi-head attention layer. Then we obtained representation vectors from local neighborhood, all nodes in graph and neighborhood in same cyclic subgraph. Combining these representations, we obtain the intermediate node representations given by:

$$\bar{X}^l = X_{MPNN,A}^l + X_{MPNN,A_C}^l + X_{MHA}^l, \quad \bar{X}^l \in R^{n \times k}$$

**Graph information layer**   The pooled graph features, extracted from the Graph Information Layer, are seamlessly integrated. Inspired by the squeeze-and-excitation block (Hu et al., 2019), this process adaptively recalibrates channel-wise graph features into each node feature as:

$$Y_{G,0}^l = \text{READOUT}\left(\{\bar{X}_v^l \mid v \in V(G)\}\right), \quad X_G^l \in R^{1 \times k}$$
$$Y_{G,1}^l = \text{ReLU}(\text{LIN}_1(Y_{G,0}^l)), \qquad\qquad Y_{G,1}^l \in R^{1 \times k/N}$$
$$Y_{G,2}^l = \text{Sigmoid}(\text{LIN}_2(Y_{G,1}^l)), \qquad\qquad Y_{G,2}^l \in R^{1 \times k}$$
$$\bar{X}_G^l = \bar{X}^l \odot Y_G^l, \qquad\qquad\qquad\quad \bar{X}_G^l \in R^{n \times k}$$

where $\text{LIN}_1$ is linear layer for squeeze feature dimension and $\text{LIN}_2$ is linear layer for excitation feature dimension. Note that $N$ is reduction factor for squeezing feature.

To culminate the process and ensure channel mixing, the features are passed through an MLP layer as follows:

$$X^l = \text{MLP}(\bar{X}_G^l), \quad X^l \in R^{n \times k}$$

### 3.3 MATHEMATICAL BACKGROUND OF TIGT

**Clique adjacency matrix**    The motivation for utilizing the clique adjacency matrix in implementing TIGT originates from the recent work by Choi et al., which establishes a mathematical identification of discerning capabilities of GNNs using the theory of covering spaces of graphs (Choi et al., 2023). To summarize their work, conventional GNNs represent two graphs $G$ and $H$, endowed with node feature functions $X_G : G \to \mathbb{R}^k$ and $X_H : G \to \mathbb{R}^k$, as identical vector representations if and only if the universal covers of $G$ and $H$ are isomorphic, and the pullback of node attributes over the universal covers are identical. We note that universal covers of graphs are infinite graphs containing unfolding trees of the graph rooted at a node as subgraphs. In other words, these universal covers do not contain cyclic subgraphs which may be found in the original given graph as subgraphs. Additional measures to further distinguish cyclic structures of graphs are hence required to boost the distinguishing power of GNNs. Among various techniques to represent cyclic subgraphs, we focus on the following two solutions which can be easily implemented: **(1)** Construct clique adjacency matrices $A_C$, as utilized in the architectural component of TIGT, which transform the geometry of universal covers themselves: **(2)** Impose additional positional encodings, which alter the pullback of node attributes over the universal covers. The distinguishing power of TIGT can be stated as follows, whose proof follows from the results shown in Choi et al. (2023).

**Theorem 3.1.** *Suppose $G$ and $H$ are two graphs with the same number of nodes and edges. Suppose that there exists a cyclic subgraph $C$ that is an element of a cycle basis of $G$ such that satisfies the following two conditions: (1) $C$ does not contain any proper cyclic subgraphs (2) Any element of a cycle basis of $H$ is not isomorphic to $C$. Then TIGT can distinguish $G$ and $H$ as non-isomorphic.*

As a corollary of the above theorem, we obtain the following explicit quantification of discriminative power of TIGT in classifying graph isomorphism classes. We leave the details of the proofs of both theorems in Appendix A.1 and A.2.

**Theorem 3.2.** *There exists a pair of graphs $G$ and $H$ such that TIGT can distinguish as non-isomorphic whereas 3-Weisfeiler-Lehman (3-WL) test cannot.*

Theorem 3.2 hence shows that TIGT has capability to distinguish pairs of graphs which are not distinguishable by algorithms comparable to 3-WL test, such as the generalized distance Weisfeiler-Lehman test (GD-WL) utilizing either shortest path distance (SPD) or resistance distance (RD) (Zhang et al., 2023).

**Graph biconnectivity**    Given that TIGT is able to distinguish classes of graphs that 3-WL cannot, it is reasonable to ask whether TIGT can capture topological properties of graphs that state-of-the-art techniques can encapsulate, which is the problem of detecting bi-connectivity of graphs (Zhang et al., 2023; Ma et al., 2023). We recall that a connected graph $G$ is vertex (or edge) biconnected if there exists a vertex $v$ (or an edge $e$) such that $G \setminus \{v\}$ (or $G \setminus \{e\}$) has more connected components than $G$. As these state-of-the-art techniques can demonstrate, TIGT as well is capable to distinguish vertex (or edge) bi-connectivity. The idea of the proof relies on comparing the Euler characteristic formula for graphs $G$ and $G \setminus \{v\}$ (or $G \setminus \{e\}$), the specific details of which are provided in Appendix A.3.

**Theorem 3.3.** *Suppose $G$ and $H$ are two graphs with the same number of nodes, edges, and connected components such that $G$ is vertex (or edge) biconnected, whereas $H$ is not. Then TIGT can distinguish $G$ and $H$ as non-isomorphic graphs.*

In fact, as shown in Appendix C of Zhang et al. (2023), there are state-of-the-art techniques which are designed to encapsulate cycle structures or subgraph patterns but cannot distinguish biconnectivity of classes of graphs, such as cellular WL (Bodnar et al., 2021a), simplicial WL (Bodnar et al., 2021b), and GNN-AK (Zhao et al., 2022). These results indicate that TIGT can detect both cyclic structures and bi-connectivity of graphs, thereby addressing the topological features the generalized construction of Weisfeiler-Lehman test aims to accomplish, as well as showing capabilities of improving distinguishing powers in comparison to other pre-existing techniques.

**Positional encoding**    As aforementioned, the method of imposing additional positional encodings to graph attributes can allow neural networks to distinguish cyclic structures, as shown in various types of Graph Transformers (Ma et al., 2023; Rampášek et al., 2023; Ying et al., 2021). One drawback, however, is that these encodings may not be effective enough to represent classes of topologically non-isomorphic graphs as distinct vectors which are not similar to one another. We present a

heuristic argument of the drawback mentioned above. Suppose we utilize a Transformer with finitely many layers to obtain vector representations $X_G^*, X_H^* \in \mathbb{R}^m$ of two graphs $G$ and $H$ with node attribute matrices $X_G, X_H \in \mathbb{R}^{n \times k}$ and positional encoding matrices $POS_G, POS_H \in \mathbb{R}^{n \times k'}$. Suppose further that all layers of the Transformer are comprised of compositions of Lipschitz continuous functions. This implies that the Transformer can be regarded as a Lipschitz continuous function from $\mathbb{R}^{n \times (k+k')}$ to $\mathbb{R}^m$. Hence, for any $\epsilon > 0$ such that $\|[X_G|POS_G] - [X_H|POS_H(v)]\| < \epsilon$, there exists a fixed constant $K > 0$ such that $\|X_G^* - X_H^*\| < K\epsilon$. This suggests that if the node attributes and the positional encodings of non-isomorphic classes of graphs are similar to one another, say within $\epsilon$-error, then such Transformers will represent these graphs as similar vectors, say within $K\epsilon$-error. Hence, it is crucial to determine whether the given positional encodings effectively perturbs the node attributes to an extent that results in obtaining markedly different vector representations.

In relation to the above observation, we show that the relative random walk probabilities positional encoding (RRWP) suggested in Ma et al. (2023) may not effectively model $K$ steps of random walks on graphs $G$ containing a cyclic subgraph with odd number of nodes and may not be distinguishable by 1-WL as $K$ grows arbitrarily large, the proof of which is outlined in Appendix A.4.

**Theorem 3.4.** *Let $\mathcal{G}$ be any collections of graphs whose elements satisfy the following three conditions: (1) All graphs $G \in \mathcal{G}$ share the same number of nodes and edges: (2) Any $G \in \mathcal{G}$ contains a cyclic subgraph with odd number of nodes: (3) For any number $d \geq 1$, all the graphs $G \in \mathcal{G}$ have identical number of nodes whose degree is equal to $d$. Fix an integer $K$, and suppose the node indices for $G \in \mathcal{G}$ are ordered based on its increasing degrees. Let $\mathbf{P}$ be the RRWP positional encoding associated to $G$ defined as $\mathbf{P}_{i,j} := [\boldsymbol{I}, \boldsymbol{M}, \boldsymbol{M}^2, \cdots, \boldsymbol{M}^{K-1}]_{i,j} \in \mathbb{R}^K$, where $\boldsymbol{M} := \boldsymbol{D}^{-1}\boldsymbol{A}$ with $\boldsymbol{A}$ being the adjacency matrix of $G$, and $\boldsymbol{D}$ the diagonal matrix comprised of node degrees of $G$. Then there exists a unique vector $\pi \in \mathbb{R}^n$ independent of the choice of elements in $\mathcal{G}$ and a number $0 < \gamma < 1$ such that for any $0 \leq l \leq K - 1$, we have $\max_{(i,j)} \|\boldsymbol{M}_{i,j}^l - \pi_j\| < \gamma^l$.*

In particular, the theorem states that the positional encodings which are intended to model $K$ steps of random walks converge at a geometric rate to a fixed encoding $\pi \in \mathbb{R}^K$ regardless of the choice of non-isomorphism classes of graphs $G \in \mathcal{G}$. Hence, such choices of positional encodings may not be effective enough to represent differences in topoloical structures among such graphs as differences in their vector representations.

## 4 EXPERIMENTS

**Dataset**    To analyze the effectiveness of TIGT compared to other models in terms of expressive powers, we experiment on the Circular Skip Link(CSL) dataset (Murphy et al., 2019). CSL dataset is comprised of graphs that have different skip lengths $R \in \{2, 3, 4, 5, 6, 9, 11, 12, 13, 16\}$ with 41 nodes that have the same features. Further, we utilize well-known graph-level benchmark datasets to evaluate proposed models compared to other models. We leverage five datasets from the "Benchmarking GNN" studies: MNIST, CIFAR10, PATTERN, and CLUSTER, adopting the same experimental settings as prior research (Dwivedi et al., 2022). Additionally, we use two datasets from the "Long-Range Graph Benchmark" (Dwivedi et al., 2023): Peptides-func and Peptides-struct. Lastly, to further verify the effectiveness of the proposed model on large datasets, we perform experiments on ZINC full dataset (Irwin et al., 2012), which is the full version of the ZINC dataset with 250K graphs and PCQM4Mv2 dataset (Hu et al., 2020) which is large-scale graph regression benchmark with 3.7M graphs. These benchmark encompass binary classification, multi-label classification, and regression tasks across a diverse range of domain characteristics. The detail of the aforementioned datasets are summarized in Appendix C.1.

**Models**    To evaluate the discriminative power of TIGT, we compare a set of previous researches related to expressive power of GNNs on CSL dataset such as Graph Transformers (GraphGPS (Rampášek et al., 2023), GRIT (Ma et al., 2023)) and other message-passing neural networks (GCN Kipf & Welling (2017), GIN, Relational Pooling GIN(RP-GIN) (Murphy et al., 2019), Cy2C-GNNs (Choi et al., 2023)). We compare our approach on well-known benchmark datasets to test graph-level test with the latest SOTA techniques, widely adopted MPNNs models, and various Graph Transformer-based studies: GRIT (Ma et al., 2023), GraphGPS (Rampášek et al., 2023)), GCN (Kipf & Welling, 2017), GIN (Xu et al., 2019), its variant with edge-features (Hu et al., 2020), GAT (Veličković et al., 2018), GatedGCN (Bresson & Laurent, 2018), GatedGCN-LSPE (Dwivedi et al., 2022), PNA (Corso et al., 2020), Graphormer (Ying et al., 2021), K-Subgraph SAT (Chen

et al., 2022), EGT (Hussain et al., 2022), SAN (Kreuzer et al., 2021), Graphormer-URPE (Luo et al., 2022), Graphormer-GD Zhang et al. (2023), DGN (Beaini et al., 2021), GSN (Bouritsas et al., 2021), CIN (Bodnar et al., 2021b), CRaW1 (Tönshoff et al., 2023), and GIN-AK+ (Zhao et al., 2022).

**TIGT Setup** For hyperparameters of models on CSL datasets, we fixed the hidden dimension and batch size to 16, and other hyperparameters were configured similarly to the setting designed for the ZINC dataset. For a fair comparison of the other nine benchmark datasets, we ensured that both the hyperparameter settings closely matched those found in the GraphGPS (Rampášek et al., 2023) and GRIT (Ma et al., 2023) studies. The differences in the number of trainable parameters between TIGT and GraphGPS primarily arise from the additional components introduced to enrich topological information within the Graph Transformer layers. Further details on hyperparameters, such as the number of layers, hidden dimensions, and the specific type of MPNNs, are elaborated upon in the Appendix C.2.

**Performance on the CSL dataset** In order to test the expressive power of the proposed model and state-of-the-art Graph Transformers, we evaluated their performance on the synthetic dataset CSL. The test performance metrics are presented in Table 1. Our analysis found that TIGT, GPS with random-walk structural encoding (RWSE), and GPS with RWSE and Laplacian eigenvectors encodings (LapPE) outperformed other models. However, the recent state-of-the-art model, GRIT with Relative Random Walk Probabilities (RRWP), could not discriminate CSL class. Interestingly, TIGT demonstrated resilience in maintaining a near 100% performance rate, irrespective of the number of added Graph Transformer layers. This consistent performance can be attributed to TIGT's unique Dual-path message-passing layer, which ceaselessly infuses topological information across various layers. Conversely, other models, which initially derive benefits from unique node attribution facilitated by positional encoding, showed signs of diminishing influence from this attribution as the number of layers grew. Additionally, we compared our findings with those of GAT and Cy2C-GNNs models. Consistent with previous studies Choi et al. (2023), GAT was unable to perform the classification task on the CSL dataset effectively. In the case of the Cy2C-GNN model, while it demonstrated high accuracy in a single-layer configuration, similar to GPS, we observed a decline in classification performance as the number of layers increased.

**Results from benchmark datasets** First, we present the test performance on five datasets from Benchmarking GNNs (Dwivedi et al., 2022) in Table 2. The mean and standard deviation are reported over four runs using different random seeds. It is evident from the results that our model ranks either first or second in performance on three benchmark datasets: ZINC, MNIST, and CIFAR10. However, for the synthetic datasets, PATTERN, and CLUSTER, our performance is found to be inferior compared to recent state-of-the-art models but is on par with the GraphGPS model. Next, we further assess the effectiveness of our current model by evaluating its test performance on four datasets from the "Long-Range Graph Benchmark" (Dwivedi et al., 2023), full ZINC dataset (Irwin et al., 2012), and the PCQM4Mv2 dataset (Hu et al., 2020). In the large datasets, the full version of the ZINC dataset and the PCQM4Mv2 dataset, TIGT consistently outperforms other models. In particular, on the PCQM4Mv2 dataset, our model demonstrated superior performance with fewer parameters compared to state-of-the-art models. In the "Long-Range Graph Benchmark," our model also present the second-highest performance compared to other models. Through all these experimental results, it is evident that by enhancing the discriminative power to differentiate isomorphisms of graphs, we can boost the predictive performances of Graph Transformers. This has enabled us to achieve competitive results in GNN research, surpassing even recent state-of-the-art model on several datasets. In a comparative analysis between Cy2C-GNN and TIGT, we observed a significant increase in performance across all datasets with TIGT. This indicates that the topological non-trivial features of graphs are well-reflected in TIGT, allowing for both a theoretical increase in expressive power and improved performance on benchmark datasets.

## 5 CONCLUSION

In this paper, we introduced TIGT, a novel Graph Transformer designed to enhance the predictive performance and expressive power of Graph Transformers. This enhancement is achieved by incorporating a topological positional embedding layer, a dual-path message passing layer, a global attention layer, and a graph information layer. Notably, our topological positional embedding layer

Table 1: Results of graph classification obtained from CSL dataset (Murphy et al., 2019). Note that a bold method indicate the results obtained by ours. All results other than the bold method are cited from available results obtained from pre-existing publications. Note that mean ± standard deviation of 4 runs with different random seeds in our results.Highlighted are the top first, second, and third results.

| GNNs | GIN | RP-GIN | GCN | Cy2C-GCN-1 |
|---|---|---|---|---|
| | 10.0±0.0 | 37.6±12.9 | 10.0±0.0 | 91.3±1.6 |
| **GATs** | GAT-1 | GAT-2 | GAT-5 | GAT-10 |
| | 10.0±0.0 | 10.0±0.0 | 10.0±0.0 | 10.0±0.0 |
| **Cy2C-GNNs** | Cy2C-GIN-1 | Cy2C-GIN-2 | Cy2C-GIN-5 | Cy2C-GIN-10 |
| | 98.33±3.33 | 46.67±38.20 | 9.17±5.69 | 7.49±3.21 |
| **GPS** | 1 layer | 2 layers | 5 layers | 10 layers |
| | 5.0±3.34 | 6.67±9.43 | 3.34±3.85 | 5.0±3.34 |
| **GPS+RWSE** | 1 layer | 2 layers | 5 layers | 10 layers |
| | 88.33±11.90 | 93.33±11.55 | 90.00±11.06 | 75.0±8.66 |
| **GPS+LapPE+RWSE** | 1 layer | 2 layers | 5 layers | 10 layers |
| | 100±0.0 | 95±10.0 | 93.33±13.33 | 86.67±10.89 |
| **GRIT+RRWP** | 1 layer | 2 layers | 5 layers | 10 layers |
| | 10.0±0.0 | 10.0±0.0 | 10.0±0.0 | 10.0±0.0 |
| **TIGT** | 1 layer | 2 layers | 5 layers | 10 layers |
| | 98.33±3.35 | 100±0.0 | 100±0.0 | 100±0.0 |

Table 2: Graph classification and regression results obtained from five benchmarks from (Dwivedi et al., 2022). Note that N/A indicate the methods which do not report test results on the given graph data set and a bold method indicate the results obtained by ours. All results other than the bold method are cited from available results obtained from pre-existing publications. Note that mean ± standard deviation of 4 runs with different random seeds in our results.Highlighted are the top first, second, and third results.

| | ZINC | MNIST | CIFAR10 | PATTERN | CLUSTER |
|---|---|---|---|---|---|
| Model | MAE↓ | Accuracy↑ | Accuracy↑ | Accuracy↑ | Accuracy↑ |
| GCN | 0.367±0.011 | 90.705±0.218 | 55.710±0.381 | 71.892±0.334 | 68.498±0.976 |
| GIN | 0.526±0.051 | 96.485±0.252 | 55.255±1.527 | 85.387±0.136 | 64.716±1.553 |
| GAT | 0.384±0.007 | 95.535±0.205 | 64.223±0.455 | 78.271±0.186 | 73.840±0.326 |
| GatedGCN | 0.282±0.015 | 97.340±0.143 | 67.312±0.311 | 85.568±0.088 | 73.840±0.326 |
| GatedGCN+LSPE | 0.090±0.001 | N/A | N/A | N/A | N/A |
| PNA | 0.188±0.004 | 97.94±0.12 | 70.35±0.63 | N/A | N/A |
| DGN | 0.168±0.003 | N/A | 72.838±0.417 | 86.680±0.034 | N/A |
| GSN | 0.101±0.010 | N/A | N/A | N/A | N/A |
| CIN | 0.079±0.006 | N/A | N/A | N/A | N/A |
| CRaW1 | 0.085±0.004 | 97.944±0.050 | 69.013±0.259 | N/A | N/A |
| GIN-AK+ | 0.080±0.001 | N/A | 72.19±0.13 | 86.850±0.057 | N/A |
| SAN | 0.139±0.006 | N/A | N/A | 86.581±0.037 | 76.691±0.65 |
| Graphormer | 0.122±0.006 | N/A | N/A | N/A | N/A |
| K-Subgraph SAT | 0.094±0.008 | N/A | N/A | 86.848±0.037 | 77.856±0.104 |
| EGT | 0.108±0.009 | 98.173±0.087 | 68.702±0.409 | 86.821±0.020 | 79.232±0.348 |
| Graphormer-GD | 0.081±0.009 | N/A | N/A | N/A | N/A |
| GPS | 0.070±0.004 | 98.051±0.126 | 72.298±0.356 | 86.685±0.059 | 78.016±0.180 |
| GRIT | 0.059±0.002 | 98.108±0.111 | 76.468±0.881 | 87.196±0.076 | 80.026±0.277 |
| **Cy2C-GNNs** | 0.102±0.002 | 97.772±0.001 | 64.285±0.005 | 86.048±0.005 | 64.932±0.003 |
| **TIGT** | 0.057±0.002 | 98.230±0.133 | 73.955±0.360 | 86.680±0.056 | 78.033±0.218 |

Table 3: Graph-level task results obtained from two long-range graph benchmarks (Dwivedi et al., 2023) , ZINC-full dataset (Irwin et al., 2012) and PCQM4Mv2 (Hu et al., 2020). Note that N/A indicate the methods which do not report test results on the given graph data set and a bold method indicate the results obtained by ours. All results other than the bold method are cited from available results obtained from pre-existing publications. Note that mean $\pm$ standard deviation of 4 runs with different random seeds in our results. Highlighted are the top first, second, and third results.

| | Long-range graph benchmark | | ZINC-full | | PCQM4Mv2 | | |
| | Peptides-func | Peptides-struct | | | | | |
| Model | AP↑ | MAE↓ | Model | MAE↓ | Model | MAE(Valid)↓ | # Param |
|---|---|---|---|---|---|---|---|
| GCN | 0.5930±0.0023 | 0.3496±0.0013 | GCN | 0.113±0.002 | GCN | 0.1379 | 2.0M |
| GINE | 0.5498±0.0079 | 0.3547±0.0045 | GIN | 0.088±0.002 | GIN | 0.1195 | 3.8M |
| GatedGCN | 0.5864±0.0035 | 0.3420±0.0013 | GAT | 0.111±0.002 | GCN-virtual | 0.1195 | 4.9M |
| GatedGCN+RWSE | 0.6069±0.0035 | 0.3357±0.0006 | SignNet | 0.024±0.003 | GIN-virtual | 0.1083 | 6.7M |
| Transformer+LapPE | 0.6326±0.0126 | 0.2529±0.016 | Graphormer | 0.052±0.005 | Graphormer | 0.0864 | 48.3M |
| SAN+LapPE | 0.6384±0.0121 | 0.2683±0.0043 | Graphormer-URPE | 0.028±0.002 | GRPE | 0.0890 | 46.2M |
| SAN+RWSE | 0.6439±0.0075 | 0.2545±0.0012 | Graphormer-GD | 0.025±0.004 | TokenGT (Lap) | 0.0910 | 48.5M |
| GPS | 0.6535±0.0041 | 0.2500±0.0012 | GPS | N/A | GPS-medium | 0.0858 | 19.4M |
| GRIT | 0.6988±0.0082 | 0.2460±0.0012 | GRIT | 0.023±0.001 | GRIT | 0.0859 | 16.6M |
| **Cy2C-GNNs** | 0.5193±0.0025 | 0.2521±0.0012 | **Cy2C-GNNs** | 0.042±0.001 | **Cy2C-GNNs** | 0.0956 | 4M |
| **TIGT** | **0.6679±0.0074** | **0.2485±0.0015** | **TIGT** | **0.014±0.001** | **TIGT** | **0.0826** | 13.0M |

is learnable and leverages MPNNs. It integrates universal covers drawn from the original graph structure and a modified structure enriched with cyclic subgraphs. This integration aids in detecting isomorphism classes. Throughout its architecture, TIGT encodes cyclic subgraphs at each layer using the dual-path message passing mechanism, ensuring that expressive power is maintained as layer depth increases. Despite a modest rise in complexity, TIGT showcases superior performance in experiments on the CSL dataset, surpassing the expressive capabilities of previous GNNs and Graph Transformers. Additionally, both mathematical justifications and empirical evaluations underscore our model's competitive advantage over contemporary Graph Transformers across diverse benchmark datasets.

While TIGT can be successfully applied to graph-level tasks, there remain avenues for future exploration. Firstly, the computational complexity is limited to $O(N^2 + N_E + N_C)$ with the number of node $N$, the number of edge $N_E$ and the number of edge in cyclic subgraphs $N_C$. Especially, due to the implementation of global attention in the Transformer, computational complexity poses challenges that we are keen to address in subsequent research. Moreover, beyond the realm of graph-level tasks, there is potential to broaden the application of TIGT into areas like node classification and link prediction. Integrating the topological characteristics inherent in TIGT with these domains might uncover more profound insights and elevate predictive accuracy.

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

# A MATHEMATICAL PROOFS

This subsection focuses on listing the mathematical background required for proving a series of theorems outlined in the main text of the paper. Throughout this subsection, we regard a graph $G := (V, E)$ as a 1-dimensional topological space endowed with the closure-finiteness weak topology (CW topology), the details of which are written in Hatcher (2002)[Chapter 0, Appendix]. In particular, we may regard $G$ as a 1-dimensional CW complex, the set of nodes of $G$ corresponds to the 0-skeleton of $G$, and the set of edges of $G$ corresponds to the 1-skeleton of $G$.

By regarding $G$ as a 1-dimensional CW complex, we are able to reinterpret the infinite unfolding tree of $G$ rooted at any choice of a node $v \in G$ as a contractible infinite 1-dimensional CW complex, also known as the universal cover of $G$.

**Definition A.1.** Given any topological space $X$, the universal cover $\pi_X : \tilde{X} \to X$ is a contractible topological space such that for any point $x \in X$, there exists an open neighborhood $U$ containing $x$ such that $\pi_X^{-1}(U)$ is a disjoint union of open neighborhoods, each of which is homeomorphic to $U$.

## A.1 PROOF OF THEOREM 3.1

The proof follows immediately from the fact that TIGT utilizes clique adjacency matrix $A_C$ (or bounded clique adjacency matrix), whose mathematical importance was explored in Theorem 3.3, Lemma 4.1, and Theorem 4.3 of Choi et al. (2023). We provide an exposition of the key ideas of the proof of the above three theorems here.

Let $G$ and $H$ be two graphs endowed with node attribute functions $f_G : V(G) \to \mathbb{R}^k$ and $f_H : V(H) \to \mathbb{R}^k$. Theorem 3.3 of Choi et al. (2023) implies that conventional GNNs can represent two graphs $G$ and $H$ as identical vector representations if and only if the following two conditions hold:

- There exists an isomorphism $\varphi : \tilde{G} \to \tilde{H}$ between two universal covers of $G$ and $H$.

- There exists an equality of pullback of node attributes $f_G \circ \pi_G = f_H \circ \pi_H \circ \varphi$.

In particular, even if $G$ and $H$ have different cycle bases whose elements consist of cyclic subgraphs not containing any other proper cyclic subgraphs, if the universal covers of $G$ and $H$ are isomorphic, then conventional GNNs cannot distinguish $G$ and $H$ as non-isomorphic.

To address this problem, one can include additional edges to cyclic subgraphs of $G$ and $H$ to alter universal covers of $G$ and $H$ to be not isomorphic to each other. This is the key insight in Lemma 4.1 of Choi et al. (2023). Any two cyclic graphs without proper cyclic subgraphs have isomorphic universal covers, both of which are homeomorphic to the real line $\mathbb{R}^1$. however, when two cyclic graphs are transformed into cliques (meaning that all the nodes lying on the cyclic graphs are connected by edges), then as long as the number of nodes forming the cyclic graphs are different, the universal covers of the two cliques are not isomorphic to one another.

The task of adjoining additional edges connecting nodes lying on a cyclic graph is executed by utilizing the clique adjacency matrix $A_C$, the matrix of which is also constructed in Choi et al. (2023). Hence, Theorem 4.3 of Choi et al. (2023) uses Lemma 4.1 to conclude that by utilizing the clique adjacency matrix $A_C$ (or the bounded clique adjacency matrix), one can add suitable edges to cyclic subgraphs of $G$ and $H$ which do not contain any proper cyclic subgraphs, thereby constructing non-isomorphic universal covers of $G$ and $H$ which allow conventional GNNs to represent $G$ and $H$ as non-identical vectors. In a similar vein, TIGT also utilizes clique adjacency matrices $A_C$ as an input data, the data of which allows one to add suitable edges to cyclic subgraphs of any classes of graphs to ensure constructions of their non-isomorphic universal covers.

## A.2 PROOF OF THEOREM 3.2

We now prove that TIGT is capable of distinguishing a pair of graphs $G$ and $H$ which are not distinguishable by 3-WL. The graphs of our interest are non-isomorphic families of strongly regular graphs $SR(16, 6, 2, 2)$, in particular the $4 \times 4$ rook's graph and the Shrikhande graph. Both graphs are proven to be not distinguishable by 3-Weisfeiler-Lehman test (Bodnar et al., 2021b)[Lemma 28], but possess different cycle bases whose elements comprise of cyclic graphs which does not contain

any proper cyclic subgraphs (Bodnar et al., 2021a)[Theorem 16]. Theorem 3.1 hence implies that TIGT is capable of distinguishing the $4 \times 4$ rook's graph and the Shrikhande graph.

We note that these types of strongly regular graphs are also utilized to demonstrate the superiority of a proposed GNN to 3-WL test, such as graph inductive bias Transformers (GRIT) (Ma et al., 2023) or cellular Weisfeiler-Lehman test (CWL) (Bodnar et al., 2021a).

### A.3 PROOF OF THEOREM 3.3

Next, we demonstrate that TIGT is also capable of distinguishing biconnectivity of pairs of graphs $G$ and $H$. Recall that the Euler characteristic formula (Hatcher, 2002)[Theorem 2.44] for graphs imply that

$$\#E(G) - \#V(G) = \# \text{ Connected components of } G - \# \text{ cycle basis of } G$$

where the term "# cycle basis of $G$" is the number of elements of a cycle basis of $G$. This number is well-defined regardless of the choice of a cycle basis, because its number is equal to the dimension of the first homology group of $G$ with rational coefficients, one of the topological invariants of $G$.

Without loss of generality, assume that $G$ is vertex-biconnected whereas $H$ is not. Then there exists a vertex $v \in V(G)$ such that $G \setminus \{v\}$ has more connected components than $G$ and $H$. This implies that given any choice of bijection $\phi : V(G) \to V(H)$ between the set of nodes of $G$ and $H$, the graphs $G \setminus \{v\}$ and $H \setminus \phi(\{v\})$ satisfy the following series of equations:

$$\# \text{ Connected components of } H \setminus \phi(\{v\}) - \# \text{ cycle basis of } H \setminus \phi(\{v\})$$
$$= \#E(H \setminus \phi(\{v\})) - \#V(H \setminus \phi(\{v\}))$$
$$= \#E(H) - \#V(H) + 1$$
$$= \#E(G) - \#V(G) + 1$$
$$= \#E(G \setminus \{v\}) - \#V(G \setminus \{v\})$$
$$= \# \text{ Connected components of } G \setminus \{v\} - \# \text{ cycle basis of } G \setminus \{v\}$$

By the condition that $G$ is vertex-biconnected whereas $H$ is not, it follows that the number of cycle basis of $G \setminus \{v\}$ and the number of cycle basis of $H \setminus \{\phi(v)\}$ are different. Because the above equations hold for any choice of cycle bases $G$ and $H$, we can further assume that both cycle bases $G$ and $H$ satisfy the condition that all elements do not contain proper cyclic subgraphs. But because the number of edges and vertices of the two graphs $G \setminus \{v\}$ and $H \setminus \{\phi(v)\}$ are identical, it follows that there exists a number $c > 0$ such that the number of elements of cycle bases of $G \setminus \{v\}$ and $H \setminus \phi(\{v\})$ whose number of nodes is equal to $c$ are different. Hence, the two graphs $G$ and $H$ can be distinguished by TIGT via the utilization of clique adjacency matrices of $G \setminus \{(v)\}$ and $H \setminus \phi(\{v\})$, i.e. applying Theorem 3.1 to two graphs $G \setminus \{(v)\}$ and $H \setminus \phi(\{v\})$.

In fact, the theorem can be generalized to distinguish any pairs of graphs $G$ and $H$ with the same number of edges, nodes, and connected components, whose number of components after removing a single vertex or an edge become different. We omit the proof of the corollary, as the proof is a direct generalization of the proof of Theorem 3.3.

**Corollary A.2.** *Let $G$ and $H$ be two graphs with the same number of nodes, edges, and connected components. Suppose there exists a pair of nodes $v \in V(G)$ and $w \in V(H)$ (or likewise a pair of edges $e_1 \in E(G)$ and $e_2 \in E(H)$) such that the number of connected components of $G \setminus \{v\}$ and $H \setminus \{w\}$ are different (and likewise for $G \setminus \{e_1\}$ and $H \setminus \{e_2\}$). Then TIGT can distinguish $G$ and $H$ as non-isomorphic graphs.*

### A.4 PROOF OF THEOREM 3.4

The idea of the proof follows from focuses on reinterpreting the probability matrix $\mathbf{M} := \mathbf{D}^{-1}\mathbf{A}$ as a Markov chain defined over a graph $G \in \mathcal{G}$.

Let's recall the three conditions applied to the classes of graphs inside our collection $\mathcal{G}$:

- All graphs $G \in \mathcal{G}$ share the same number of nodes and edges
- Any $G \in \mathcal{G}$ contains a cyclic subgraph with odd number of nodes

- For any number $d \geq 1$, all graphs $G \in \mathcal{G}$ have identical number of nodes whose degree is equal to $d$.

Denote by $n$ the number of nodes of any graph $G \in \mathcal{G}$. The second condition implies that any graph $G \in \mathcal{G}$ is non-bipartite, hence the probability matrix $\mathbf{M}$ is an irreducible aperiodic Markov chain over the graph $G$. In particular, this shows that the Markov chain $\mathbf{M}$ has a unique stationary distribution $\pi \in \mathbb{R}^n$ such that the component of $\pi$ at the $j$-th node of $G$ satisfies

$$\pi_j = \frac{d(j)}{2 \# E(G)}$$

where $d(j)$ is the degree of the node $j$ (Lovasz, 1993)[Section 1]. The first condition implies that regardless of the choice of the graph $G \in \mathcal{G}$, the stationary distributions of $\pi$ obtained from such Markov chains associated to each $G$ are all identical up to re-ordering of node indices based on their node degrees. The geometric ergodicity of Markov chains, as stated in Lovasz (1993)[Theorem 5.1, Corollary 5.2], show that for any initial probability distribution $\delta \in \mathbb{R}^n$ over the graph $G$, there exists a fixed constant $C > 0$ such that for any $l \geq 0$,

$$\max_j |(\delta^T \mathbf{M}^l)_j - \pi_j| < C \times \gamma^l$$

The geometric rate of convergence $\gamma$ satisfies the inequality $0 < \gamma < 1$. We note that the value of $\gamma$ is determined from eigenvalues of the matrix $N := D^{-1/2} \mathbf{M} D^{1/2}$, all of whose eigenvalues excluding the largest eigenvalue is known to have absolute values between $0$ and $1$ for non-bipartite graphs $G$ (Lovasz, 1993)[Section 3]. To obtain the statement of the theorem, we apply (**??**) with probability distributions $\delta$ whose $i$-th component is 1, and all other components are equal to 0.

## B  ABALATION STUDY

To understand the significance of each component in our deep learning model, we performed multiple ablation studies using the ZINC dataset (Dwivedi et al., 2022). The results are presented in Table 4. The influence of the graph information and the topological positional embedding layer is relatively marginal compared to other layers. The choice of weight-sharing within the topological positional embedding layer, as well as the selection between the hyperbolic tangent and ReLU activation functions, play a significant role in the model's performance. Likewise, opting for Single-path MPNNs, excluding the adjacency matrix instead of the proposed Dual-path in each TIGT layer, results in a considerable performance drop. Within the graph information layer, it's evident that employing a sum-based readout function, akin to graph pooling, is crucial for extracting comprehensive graph information and ensuring optimal results. Additionally, we experimented with applying the Performer, which utilizes a kernel trick to replace the quadratic complexity of the transformer's global attention with linear complexity, in our TIGT model. However, we found that this resulted in performance similar to models that did not use global attention. This suggests that TIGT may require further research to address the issue of quadratic complexity effectively. In a similar setting, we conducted experiments with Cy2C-GNN, which has fewer parameters (114,433) compared to TIGT, and observed poorer performance. We also tested a larger version of Cy2C-GNN, named Cy2C-GNN(Large), with 1,766,401 parameters—approximately three times more than TIGT's 539,873—only to find that this resulted in a worse mean absolute error (MAE).

## C  IMPLEMENTATION DETAILS

### C.1  DATASETS

A detail of statistical properties of benchmark datasets are summarized in Table 5. We perform the experiments on GraphGPS Rampášek et al. (2023) framework.

### C.2  HYPERPARAMETERS

For all models we tested on the CSL dataset, we consistently set the hidden dimension to 32 and the batch size to 5. Other hyperparameters were kept consistent with those used for the models evaluated on the zinc dataset.

Table 4: Ablation study to analyze the effectiveness of the component of TIGT on the ZINC dataset. (Dwivedi et al., 2022).

| ZINC | MAE↓ |
|---|---|
| **TIGT** | 0.057±0.002 |
| w/o graph information | 0.059±0.005 |
| w/o topological positional embedding | 0.060±0.003 |
| not share weight in topological positional embedding | 0.061±0.003 |
| Transformer → Performer in global attention | 0.063±0.001 |
| w/o global attention | 0.063±0.003 |
| Tanh → ReLU in topological positional embedding | 0.063±0.004 |
| Dual-path MPNNs → Single-path MPNNs | 0.064±0.003 |
| Sum → Mean readout in graph information | 0.069±0.001 |
| Cy2C-GNNs | 0.102±0.002 |
| Cy2C-GNNs(Large) | 0.121±0.003 |
| GraphGPS | 0.070±0.004 |

Table 5: Summary of the statistics of dataset in overall experiments (Dwivedi et al., 2022; 2023; Irwin et al., 2012; Hu et al., 2020).

| Dataset | ZINC/ZINC-full | MNIST | CIFAR10 | PATTERN | CLUSTER | Peptides-func | Peptides-struct | PCQM4Mv2 |
|---|---|---|---|---|---|---|---|---|
| # Graphs | 12,000/250,000 | 70,000 | 60,000 | 14,000 | 12,000 | 15,535 | 15,535 | 3,746,620 |
| Average # nodes | 23.2 | 70.6 | 117.6 | 118.9 | 117.2 | 150.9 | 150.9 | 14.1 |
| Average # edges | 24.9 | 564.5 | 941.1 | 3,039.3 | 2,150.9 | 307.3 | 307.3 | 14.6 |
| Directed | No | Yes | Yes | No | No | No | No | No |
| Prediction level | Graph | Graph | Graph | Inductive node | Inductive node | Graph | Graph | Graph |
| Task | Regression | 10-class classfi. | 10-class classfi. | Binary classif. | 6-class classif. | 10-task classif. | 11-task regression | Regression |
| Metric | Mean Abs. Error | Accuracy | Accuracy | Weighted Accuracy | Accuracy | Avg. Precision | Mean Abs. Error | Mean Abs. Error |
| Average # H1 cycles | 2.8/2.8 | 212.1 | 352.5 | 2921.4 | 2034 | 3.7 | 3.7 | 1.4 |
| Average magnitude # cycles | 5.6/5.6 | 4.4 | 5.1 | 3.6 | 4.1 | 6.7 | 6.7 | 4.9 |
| # graph w/o cycles | 66/1109 | 0 | 0 | 0 | 0 | 1408 | 1408 | 444736 |

To ensure a fair comparison, we followed the hyperparameter settings of GraphGPS (Rampášek et al., 2023) as outlined in their benchmark datasets. It's worth noting that, due to the intrinsic nature of the TIGT architecture, the number of model parameters varies. Details regarding these hyperparameters are provided in Table 6.

## C.3 IMPLEMENTATION DETAIL OF EXPERIMENT ON CSL DATASET

The CSL dataset Murphy et al. (2019) was obtained using the 'GNNBenchmarkDataset' option from the PyTorch Geometric library Fey & Lenssen (2019). We partitioned the dataset into training, validation, and test sets with proportions of 0.6, 0.2, and 0.2, respectively. Detailed descriptions of the hyperparameters are presented in Table 7. Hyperparameters for the CSL dataset that are not

Table 6: Hyperparameters for ten datasets from BenchmarkingGNNs Dwivedi et al. (2022), ZINC-full Irwin et al. (2012), the Long-range Graph Benchmark Dwivedi et al. (2023) and PCQM4Mv2 Hu et al. (2020).

| Layer | | ZINC/ZINC-full | MNIST | CIFAR10 | PATTERN | CLUSTER | Peptide-func | Peptides-struct | PCQM4Mv2 |
|---|---|---|---|---|---|---|---|---|---|
| | MPNNs | GIN | GatedGCN | GAT | GatedGCN | GIN | GIN | GIN | GIN |
| | Weights of MPNNs | Share | Not share | Share | Not share | Not share | Not share | Not share | Not share |
| Topological P.E | Activation | Tanh | Tanh | Tanh | ReLU | Tanh | Tanh | ReLU | ReLU |
| | Normalize | Batch | Batch | Batch | Batch | Batch | Batch | Batch | Batch |
| | Self-loop | False | False | False | False | False | False | False | True |
| | MPNNs | GIN | GatedGCN | GAT | GatedGCN | GatedGCN | GatedGCN | GIN | GatedGCN |
| Dual-path MPNNs | Weights of MPNNs | Not share | Not share | Single-path | Single-path | Single-path | Single-path | Single-path | Not share |
| | Dropout | 0.0 | 0.0 | 0.05 | 0.05 | 0.05 | 0.0 | 0.05 | 0.05 |
| | # Layers | 10 | 3 | 3 | 4 | 6 | 4 | 4 | 10 |
| Global attention | Hidden dim | 64 | 52 | 52 | 64 | 48 | 96 | 96 | 256 |
| | # Heads | 4 | 4 | 4 | 8 | 8 | 4 | 4 | 8 |
| | Attention dropout | 0.5 | 0.5 | 0.8 | 0.2 | 0.8 | 0.5 | 0.5 | 0.2 |
| | Residual connection | True(In) | False | True(In) | True(In) | True(In) | True | True | True |
| Graph information | Pooling | Sum | Mean | Mean | Sum | Mean | Sum | Sum | Mean |
| | Reduction factor | 4 | 4 | 4 | 4 | 4 | 4 | 4 | 4 |
| Graph pooling | | Sum | Mean | Mean | - | - | Mean | Mean | Mean |
| | Batch size | 32/256 | 16 | 16 | 32 | 16 | 32 | 32 | 256 |
| Train | Learning rate | 0.001 | 0.001 | 0.001 | 0.0005 | 0.001 | 0.0003 | 0.0003 | 0.0005 |
| | # Epochs | 2000 | 200 | 100 | 100 | 100 | 200 | 200 | 250 |
| | # Weight decay | 1e-5 | 1e-5 | 1e-5 | 1e-5 | 1e-5 | 0.0 | 0.0 | 0.0 |
| # Parameters | | 539873 | 190473 | 98381 | 279489 | 533814 | 565066 | 574475 | 13.0M |

Table 7: Hyperparameters for ten datasets from CSL dataset.

| Layer | | Cy2C-GNNs | GPS+LapPE+RWSE | GRIT+RRWP | TIGT |
|---|---|---|---|---|---|
| Encoder | Type of MPNNs | GIN | GIN | - | GIN |
| | Type of Attention layer | - | Transformer | GRIT | Transformer |
| | Hidden dim | 64 | 64 | 64 | 64 |
| | # Heads | - | 4 | 4 | 4 |
| | Dropout | 0.0 | 0.0 | 0.0 | 0.0 |
| Train | Batch size | 4 | 4 | 4 | 4 |
| | Learning rate | 0.001 | 0.001 | 0.001 | 0.001 |
| | # Epochs | 200 | 200 | 200 | 200 |
| | # Weight decay | 1e-5 | 1e-5 | 1e-5 | 1e-5 |
| | # layer (prediction head) | 1 | 1 | 1 | 1 |
| Preprocessing time | | 0.24s | 0.30s | 0.11s | 0.24s |
| # Parameters/Computation time(epoch) | | | | | |
| # Layers | 1 | 36634/3.0s | 45502/4.4s | 50458/5.37s | 64490/4.2s |
| # Layers | 2 | 45082/3.3s | 87422/5.7s | 97626/5.73s | 117114/5.8s |
| # Layers | 5 | 70426/3.8s | 213182/9.4s | 236506/9.9s | 274986/10.8s |
| # Layers | 10 | 112666/5s | 422782/15.5s | 474970/13.7s | 538106/18.3s |

specified here are consistent with those used in the ZINC dataset experiment Rampášek et al. (2023); Ma et al. (2023).

