# OpenReview forum: "Topology-Informed Graph Transformer"
_ICLR.cc/2024/Conference — Submitted to ICLR 2024_

### Official Review · Reviewer_Cu2Y · 2023-10-24

**Soundness:** 2 fair
**Presentation:** 1 poor
**Contribution:** 2 fair
**Rating:** 3
**Confidence:** 4

**Summary:**

Current graph transformers lack the ability to distinguish isomorphisms of graphs, thus affecting the predictive performance of the methods. To address this problem, this paper proposes Topology-Informed Graph Transformer (TIGT). TIGT contains four components: a topological positional embedding layer, a dual-path message-passing layer, a global attention mechanism, and a graph information layer. Also, a mathematical analysis of the discriminatory ability of TIGT is given in this paper.

**Strengths:**

1.This paper is well structured.
2.In this paper, experiments are conducted on several datasets and a rich theoretical proof is given.

**Weaknesses:**

1.Lack of review of related work and comparison of TIGT with other work.
2.A graph is vertex-biconnected if it is connected and does not have any cut vertex. The definition of vertex biconnected in this paper is wrong.
3.The case used in Appendix A.2 for the proof of Theorem 3.2 is too particularized, so that the generalization of the theory and the proof cannot be guaranteed.
4.Missing ablation of the component “global attention layer” in ablation study.
5.The performance improvement of TIGT is not significant and in many cases it is not as good as GRIT.

**Questions:**

1.Can you describe the main innovations of your method compared to previous graph transformer methods?
2.The theoretical proof of Theorem 3.1 in Appendix A.1 is highly similar to the reference[1], can you specify the innovation of the theory in this paper?
3.Are the values shown in Table 1 the F1 value? How are the results obtained for the TIGT achieve 100% with variance 0.0? How are the results obtained for GRIT+RRWP?

References:
[1]Yun Young Choi, Sun Woo Park, Youngho Woo, and U Jin Choi. Cycle to clique (cy2c) graph
neural network: A sight to see beyond neighborhood aggregation. In The Eleventh International
Conference on Learning Representations, 2023.

---

> ### Author Response · Authors · 2023-11-21
> **Response to Reviewer Cu2Y**
>
> We thank the reviewer for providing helpful comments to our manuscript. Please find our responses to the weaknesses of the manuscript and questions pointed out by the reviewer.
>
> - Weakness 2: We thank the reviewer for pointing out the issue. We added a condition that the graph G is a connected graph. Nevertheless, we would like to note that the given definition of biconnectivity of graphs, assuming G is connected, is correct. The vertex v and edge e of the graph G such that the number of components of G \ {v} and G \ {e} is greater than 1 correspond to a cut vertex and a cut edge of the graph G.
> - Weakness 3: Theorem 3.2 shows that there exists a pair of graphs such that TIGT can distinguish whereas 3-WL cannot distinguish. The theorem does not claim that all graphs not distinguishable by 3-WL can be distinguished by TIGT. Hence, the current proof of Theorem 3.2 in the appendix of the manuscript is correct, because it presents a particula pair of graphs that 3-WL cannot distinguish whereas TIGT can.
> - Weakness 1, Question 1:  One prominent novelty of this work comes from obtaining an optimal balance among encapsulating the merits of Cy2C-GNNs in incorporating topological invariants of graph datasets, utilizing the advantages of Graph Transformers in effectively analyzing node and edge features of graph datasets, and sustaining the expressive power of GNNs even under increases in number of hidden layers. Our newly added implementations demonstrate that TIGT successfully implements the optimal balance of the aforementioned three conditions, whereas other GNNS or transformer architectures may not effectively satisfy all three conditions simultaneously.
> - Question 2: The theoretical innovation of the paper focuses on the capability to incorporate cyclic substructures of graphs to graph transformers, and demonstrate how cyclic substructures are related to geometric properties of graphs analyzed from previous studies. We note that Cy2C-GNN does not incorporate cyclic substructure to graph transformers. As such, Theorem 3.1 and 3.2 demonstrates that cyclic substructures can be also incorporated into graph transformer architecture, the proof of which utilizes the theoretical results from Choi et al. (2023). The manuscript also proves new theoretical results which relates cyclic substructures of graphs with previously studied innate properties of graphs (graph biconnectivity), and analyzes limitations of previously studied techniques which utilize positional encoding of graph datasets. In particular, Theorem 3.3 utilizes Euler characteristics of graphs to show that biconnectivity of graphs can be detected from analyzing cyclic substructures of graphs. Theorem 3.4 utilizes geometric ergodicity of Markov chains (i.e. that the rate that a probability distribution converges to a stationary distribution of a Markov chain is comparable to the rate that a geometric sequence converges to a limit) to show potential limitations in attempting to distinguish geometric properties of graphs without directly utilizing the cyclic substructures of graphs.
> - Question 3: The results obtained from Table 1 consist of accuracies in classifying graph datasets from CSL dataset. We note that the standard deviations are obtained from 4 runs of graph classification with different random seeds. The results from Table 1 show that TIGT is able to fully distinguish cyclic graph datasets presented in CSL dataset. For further information on implementation with CSL dataset, we added Table 7 and Appendix C.3 to the newly updated manuscript, along with supplementary submission which contains a configuration file along with sample experimental results. Using the configuration file, one can implement the graph classification using the code provided on GRIT Github repository. (Here, we set the edge and node features to be all equal to 1). Along with GRIT, we also implemented classifying CSL datasets using GAT and Cy2C-GNNs, the results of which can be observed from Table 1. We find that TIGT outperforms all the baseline models in encapsulating cyclic substructures of graphs in CSL datasets.

---

### Official Review · Reviewer_apGs · 2023-10-30

**Soundness:** 3 good
**Presentation:** 3 good
**Contribution:** 2 fair
**Rating:** 5
**Confidence:** 3

**Summary:**

This paper introduces the Topology-Informed Graph Transformer (TIGT), a novel graph transformer architecture designed to improve the discriminative capability for detecting graph isomorphisms. The TIGT model consists of four key components: a topological positional embedding using cyclic subgraphs of graphs, a dual-path message-passing layer, a global attention mechanism, and a graph information layer. Experimental results on various graph classification tasks illustrate the effectiveness of the proposed method.

**Strengths:**

- The writing of this paper is easy to follow.
- The problem of strengthening the discriminative power of distinguishing isomorphisms of graphs is crucial.
- This paper conducts extensive experiments.

**Weaknesses:**

- Originality/Novelty: The main theoretical results (Theorem 3.1 and Theorem 3.2) are mainly based on previous work (Choi et al., 2023).
- The assumptions made in Theorem 3.3, which requires two graphs to have the same number of nodes and edges, and Theorem 3.4, which assumes that all graphs have the same number of nodes with degree $d$, may have limitations in practical applications.
- It would be valuable if the authors could discuss the presence of cyclic structures in graph datasets and its impact on the proposed architecture's performance, whether it is missing or not.
- Ablation study: The authors have provided an ablation study for key components of the method. However, the topological positional embedding layer, the main contribution of this paper, appears to have a marginal effect.
- Although the authors have provided an analysis of the computational complexity for the proposed method, they have not provided empirical results regarding running time compared to baselines. I would suggest the authors also measure the running time of the method and other baselines.
- Baselines: Some baselines on graph transformer are either missing or not adequately discussed:
1. Kong, Kezhi, et al. "GOAT: A Global Transformer on Large-scale Graphs." ICML 2023.
2. Zhang, Zaixi, et al. "Hierarchical graph transformer with adaptive node sampling." NeurIPS 2022.

**Questions:**

See weaknesses above

---

> ### Author Response · Authors · 2023-11-21
> **Response to Reviewer apGs**
>
> We thank the reviewer for pointing out constructive comments which helped us improve the manuscript. Please find our response to the weaknesses of the manuscript pointed out by the reviewer.
>
> - Weakness 1: While we utilized the results from Choi et al. (2023) to prove the theoretical effectiveness in incorporating cyclic structures of graphs, one prominent novelty of this work comes from obtaining an optimal balance among encapsulating the merits of Cy2C-GNNs in incorporating topological invariants of graph datasets, utilizing the advantages of Graph Transformers in effectively analyzing node and edge features of graph datasets, and sustaining the expressive power of GNNs even under increases in number of hidden layers. Our newly added implementations demonstrate that TIGT successfully implements the optimal balance of the aforementioned three conditions. Furthermore, we also added new theoretical results which relates cyclic substructures of graphs with previously studied innate properties of graphs (graph biconnectivity), and analyzes limitations of previously studied techniques which utilize positional encoding of graph datasets.
> - Weakness 2: The conditions on pairs of graphs indicated in Theorems 3.3 and 3.4 correspond to pairs of graphs that conventional GNN may not be able to distinguish as non-isomorphic pairs. In fact, if it is the case that any pairs of two graphs does not satisfy the conditions in Theorems 3.3 and 3.4, then conventional GNNs are able to distinguish two graphs as non-isomorphic. As an example, suppose G and H are two graphs whose number of nodes are different but the number of edges are identical. Then there exists a pair of nodes v of G and w of H such that the degree of v and degree of w are not equal to each other. This implies that the universal covers of G and H are not isomorphic, in particular at the nodes of the universal covers of G and H which, respectively, project to nodes v and w. Theorem 3.3 shows that biconnectivity of graphs can be detected from analyzing cyclic substructures of graphs, and Theorem 3.4 shows potential limitations in attempting to distinguish geometric properties of graphs without directly utilizing the cyclic substructures of graphs.
> - Weakness 3: As the reviewer suggested, in Table 5 of the updated manuscript, we added a summary of cyclic substructures inherent in the graph datasets utilized in our experiments.
> - Weakness 4: In Tables 1 through 4, we added new implementations of graph classifications and graph regressions by utilizing GATs and Cy2C-GNNs. The empirical results obtained in these tables demonstrate that TIGT outperforms both techniques in both classification and regression experiments. Unlike Cy2C-GNNs, TIGT effectively incorporates clique adjacency matrices, global attention layers, and graph information layers, the architectural design of which contributes to outperformance in graph classifications and regressions.
> - Weakness 5: In Table 7, we added specifications of hyperparameters and computation time taken in classifying graph datasets from utilizing TIGT and baseline techniques.
> - Weakness 6: We thank the reviewer for suggesting new references. We realized that the new references mostly focus on node classification of graph datasets, the results of which are difficult to make direct comparisons to TIGT which focuses primarily on graph classifications. Nevertheless, we added the new references to the introduction of the manuscript.

---

### Official Review · Reviewer_8s8z · 2023-10-31

**Soundness:** 4 excellent
**Presentation:** 3 good
**Contribution:** 3 good
**Rating:** 5
**Confidence:** 4

**Summary:**

The authors have introduced an innovative Graph Transformer designed to enhance its discriminative capabilities. This paper demonstrates the ability of the proposed method to effectively distinguish graph isomorphisms through a novel dual-path message-passing layer. Both experimental and theoretical findings substantiate the authors' assertions. Moreover, the study delves into a novel positional embedding layer, aiming to harness topological information more efficiently within the Graph Transformer framework. Experimental evaluations further underscore the method's proficiency in graph-level benchmarks.

**Strengths:**

1. **Performance.** The introduced methodology exhibits outstanding performance, notably excelling on the CSL dataset and outperforming the expressive power of contemporary Graph Transformers.
2. **Theoretical Development.** Comprehensive theoretical exploration confirms that the proposed approach encompasses and advances beyond current graph transfomers.

**Weaknesses:**

1. **Novelty.** This work is merely a simple adoption of Cy2C-GNNs with multi-head attention to package that as a graph transformer. The authors present the same layer, except for edge features E^{l-1}, as two different layers in Section 3.1 and Section 3.2. They are called Topological positional embedding layer and Dual-path MPNNs, respectively. After getting node features, then some multi-head attention-based graph transformer.
2. **Scalability.** The authors discuss the scalability of the proposed method regarding the number of nodes, edges, and edges in cyclic subgraphs. The method has a quadratic complexity in terms of the number of nodes. This indicates that the method does not scale well, and it might be difficult to apply it to large-scale graphs.
3. **Applicability.** The authors showed the effectiveness of the proposed methods only on graph-level benchmarks. As the authors mentioned, the demonstration in other-level tasks, such as node classification/clustering, link prediction, and community detection, is needed to check the applicability.

**Questions:**

1. Provide more baselines, including GNNs. Especially the proposed method is very similar to Cy2c-GNN and the key module is from that paper, but the authors did not compare their method with Cy2-GNN in Table 1, where the proposed method show significant improvement against graph transformers. In Table 2 and 3, Cy2-GNNs are missing.
2. Please provide more implementation details of Cy2-GNN-1 and compare its computational cost and model [parameters.Is](http://parameters.Is) it fair to compare between TIGT and Cy2-GNN-1.

---

> ### Author Response · Authors · 2023-11-21
> **Response to Reviewer 8s8z**
>
> We thank the reviewer for kindly providing constructive feedbacks and comments to the manuscript. Please find our responses to some of the questions in the list provided below.
>
> - Weakness 1: One prominent novelty of this work comes from obtaining an optimal balance among encapsulating the merits of Cy2C-GNNs in incorporating topological invariants of graph datasets, utilizing the advantages of Graph Transformers in effectively analyzing node and edge features of graph datasets, and sustaining the expressive power of GNNs even under increases in number of hidden layers. Our newly added implementations demonstrate that TIGT successfully implements the optimal balance of the aforementioned three conditions, whereas other GNNS or transformer architectures may not effectively satisfy all three conditions simultaneously.
> - Question 1: In Table 1 of the updated manuscript, we added new comparisons of performances of TIGT to other baseline GNNs and transformers. In particular, we added new implementations of classifying graph datasets by utilizing GATs and Cy2C-GNNs. We observe that while Cy2C-GNN performs well with a single hidden layer, its performance in classifying graph datasets decreases as the number of layer increases. In addition, additional implementations demonstrate that GATs cannot effectively distinguish cyclic substructure of graph datasets.
> - Weakness 2, Question 2: In Table 7, we added a comparison of the pre-processing computation time, computation time per epoch, and the number of hyperparameters utilized in classifying graph datasets by using Cy2C-GNNs, GPS+LapPE+RWSE, GRIT+RRWP, and TIGT. In Table 4, we also added the empiricial results obtained from Cy2C-GNNs with increased number of hyperparameters. While TIGT exhibits slightly increased computational costs and hyperparameters because it is comprised of various architectural components, we observe that TIGT exhibits meaningful performance boosts in graph classifications and regressions.

---

### Author Response · Authors · 2023-11-21
**Updates**

We thank the reviewers for investing their invaluable time and effort in reading through the manuscript and giving constructive comments. Here is a list of additional improvements we made in the manuscript we submitted.

- Table 1: We added new implementations of classifying graph datasets by utilizing GATs and Cy2C-GNNs. Consistent with previous studies, we observe that  Cy2C-GNN exhibits optimal performance when the number of layers is equal to 1, but its performance declines drastically as the number of layer increases.
- Table 2, 3: We added new implementations of graph classification and regression results obtained from utilizing Cy2C-GNNs. We observe that TIGT overpowers Cy2C-GNN in effectively analyzing these graph datasets, thus suggesting how TIGT effectively incorporates transformer architecture with cyclic substructures of graphs.
- Table 4: We enriched our ablation studies using the ZINC dataset by analyzing the dataset with Performers, without global attention, and Cy2C-GNNs. While the additional ablation studies suggest that Performes without global attention exhibits comparable performance to TIGT, we observe that TIGT still greatly surpasses the mean absolute errors obtained from Cy2C-GNNs.
- Table 5: We added a summary of cyclic substructures inherent in ZINC, MNIST, CIFAR10, PATTERN, CLUSTER, Peptides-func, and Peptides-struct datasets.
- Table 7: We added specifications of hyperparameters, computational costs, and preprocessing time used in classifying graph datasets by using Cy2C-GNNs, GPS+LapPE+RWSE, GRIT+RRWP, and TIGT.

---

### Meta-Review · Area_Chair_v8Jj · 2023-12-11

**Metareview:**

This paper proposes a topology-informed graph transformer architecture which basically combines the Cycle-to-clique graph neural network (Cy2c GNN) with the global attention layer of a transformer, which has enhanced discriminative power over previous graph transformers. The authors provide theoretical analyses which show that the cyclic substructures can be incorporated into graph transformer architecture, relate cyclic substructures of graph with graph biconnectivity, and demonstrate the potential limitations of previous positional encoding techniques. They further perform extensive experimental validation of the proposed graph transformer framework on multiple benchmark datasets against both graph neural networks and graph transformers, to verify its effectiveness.

The reviewers acknowledge that the problem of enhancing the discriminative power of graph transformers is important and found the proposed theoretical analyses as well as the experimental results as adequately showing the effectiveness of the proposed framework. They also found the paper well-structured and written.

However, all reviewers leaned toward rejection based on the limited novelty and contribution over Cy2C GNNs [Choi et al. 23], as the proposed framework simply combines it with a global attention layer, and the theoretical analyses are based on the analyses from [Choi et al. 23] as well. They were also concerned with the oversimplified assumption in theoretical analyses, lack of experimental results on node-level tasks such as node classification, and limited scalability.

The authors provided responses to those comments saying that the proposed framework provides the optimal combination of the two different frameworks, and the theoretical analyses are sufficiently meaningful. However, the reviewers remained unconvinced about the novelty of the work and many of the comments such as the lack of experimental results on node classification remained unanswered by the end of the discussion period. Therefore, the paper may benefit from another round of revision, with more convincing arguments on the novelty and contribution of the work.

**Justification For Why Not Higher Score:**

Despite its effectiveness, the proposed framework looks incremental over the work of [Choi et al. 23] which propose Cy2C GNNs, and the authors did not provide convincing arguments on the novelty of the work, even in the final rebuttal.

**Justification For Why Not Lower Score:**

N/A

---

### Decision · Program_Chairs · 2024-01-16

Reject